# Analysis of the *BDNF* Gene rs6265 Polymorphism in a Group of Women with Alcohol Use Disorder, Taking into Account Personality Traits

**DOI:** 10.3390/ijms25126448

**Published:** 2024-06-11

**Authors:** Agnieszka Boroń, Aleksandra Suchanecka, Krzysztof Chmielowiec, Jolanta Chmielowiec, Jolanta Masiak, Grzegorz Trybek, Aleksandra Strońska-Pluta, Monika Rychel, Anna Grzywacz

**Affiliations:** 1Department of Clinical and Molecular Biochemistry, Pomeranian Medical University in Szczecin, Powstańców Wielkopolskich 72 St., 70-111 Szczecin, Poland; agnieszka.boron@pum.edu.pl; 2Independent Laboratory of Behavioral Genetics and Epigenetics, Pomeranian Medical University in Szczecin, Powstańców Wielkopolskich 72 St., 70-111 Szczecin, Poland; aleksandra.suchanecka@pum.edu.pl (A.S.); aleksandra.stronska@pum.edu.pl (A.S.-P.); 3Department of Hygiene and Epidemiology, Collegium Medicum, University of Zielona Góra, 28 Zyty St., 65-046 Zielona Góra, Poland; chmiele@vp.pl (K.C.); chmiele1@o2.pl (J.C.); 4II Department of Psychiatry and Psychiatric Rehabilitation, Medical University of Lublin, 1 Głuska St., 20-059 Lublin, Poland; jolantamasiak@wp.pl; 5Department of Oral Surgery, Pomeranian Medical University in Szczecin, Powstańców Wielkopolskich 72 St., 70-111 Szczecin, Poland; g.trybek@gmail.com; 6Maxillofacial Surgery Clinic, 4th Military Clinical Hospital in Wroclaw, ul. Rudolfa Weigla 5, 50-981 Wroclaw, Poland; 7Student Scientific Club of Department of Clinical and Molecular Biochemistry, Pomeranian Medical University in Szczecin, Powstańców Wielkopolskich 72 St., 70-111 Szczecin, Poland; 68069@student.pum.edu.pl

**Keywords:** BDNF, alcohol use disorder, alcohol dependency, alcohol, women

## Abstract

It seems that BDNF has a direct influence on the brain pathways and is typically engaged during the processing of rewards. A surge in BDNF levels in the ventral tegmental area (the region from which the dopaminergic neurons of the mesocorticolimbic dopamine system originate and extend to the dorsolateral and ventromedial striatum) triggers a state of reward similar to that produced by opiates in animal studies. The aims of the study were (1) to analyze the association of the *BDNF* gene rs6265 polymorphism with AUD (alcohol use disorder) in women, (2) analyze personality and anxiety in alcohol-dependent and control woman, and (3) conduct an interaction analysis of rs6265 on personality, anxiety, and alcohol dependence. Our study found a notable interaction between the anxiety (trait and state), neuroticism, rs6265, and AUD. The alcohol AUD G/A genotype carriers revealed higher level of the anxiety trait (*p* < 0.0001) and neuroticism (*p* < 0.0001) compared to the control group with G/A and G/G genotypes. The alcohol use disorder subjects with the G/A genotype displayed higher levels of an anxiety state than the control group with G/A (*p* < 0.0001) and G/G (*p* = 0.0014) genotypes. Additionally, the alcohol use disorder subjects with the G/G genotype obtained lower levels of agreeability compared to the controls with G/A (*p* < 0.0001) and G/G (*p* < 0.0001) genotypes. Our study indicates that anxiety (trait and state) and neuroticism are interacting with the *BDNF* gene rs6265 polymorphism in alcohol-dependent women. Characteristics like anxiety (both as a trait and a state) and neuroticism could have a significant impact on the mechanism of substance dependency, particularly in females who are genetically susceptible. This is regardless of the reward system that is implicated in the emotional disruptions accompanying anxiety and depression.

## 1. Introduction

According to the World Health Organization, the consumption of alcohol results in 3 million deaths each year globally and causes disability and sickness for millions more. The misuse of alcohol accounts for 5.1% of the worldwide burden of disease. For those aged 15 to 49, alcohol poses a substantial risk, leading to premature death and disability and is responsible for 10% of all deaths within this age group [1]. Over the years, men have historically had higher rates of alcohol use disorder (AUD) than women [2]. However, the gender gap is now shrinking. Between 2009 and 2019, AUD rates increased by 85% in women, while in men, they increased by only 35% [3]. The onset of the COVID-19 pandemic in 2020 accelerated the increase in the number of AUD diagnoses in women [4,5,6], underscoring the pressing need to understand the gender-specific psychosocial and neurobiological factors that drive the development and progression of AUD.

Research shows that women and men differ in terms of the risk of developing alcohol addiction/dependency and the health consequences associated with it [7]. For example, some types of stressors, such as sexual trauma, are more likely to be experienced by women than by men [8], and higher levels of stress have been shown to increase alcohol abuse and susceptibility to alcohol addiction [9]. Additionally, women demonstrate a significantly “faster and more risk-oriented pathway to compulsive drug seeking” [10]. Therefore, there is a significant need to understand gender differences in the risk of developing and maintaining alcohol use disorder (AUD) to develop new approaches to the prevention and treatment of AUD in women [11].

Substance use disorders have been characterized as diseases of maladaptive plasticity [12,13,14], and thus, neurotrophic factors represent putative molecular mediators of the long-term effects of drug abuse, including alcohol [15]. An interaction between neurotrophic factor function and alcohol was first suggested by reduced hippocampal neurotrophic activity, assessed as the ability of hippocampal extracts to support the survival of dorsal root ganglia cultures after chronic high alcohol consumption [16].

In vivo, neuroimaging studies and post-mortem studies of patients and experiments in rodents have shown that chronic excessive alcohol consumption is accompanied by atrophy and degeneration at the cellular and macrostructural levels [17]. Complex neuroadaptation is responsible for alcohol-related phenomena such as sensitization, tolerance, and withdrawal [18]. Chronic alcohol poisoning often causes affective and cognitive disorders [18,19], which are believed to be the result of abnormal neuroplasticity [19,20,21].

Many studies have focused on examining differences between peripheral BDNF levels in clinical and community samples with and without alcohol dependence [22,23,24]. They showed mixed results regarding the association between BDNF levels and the diagnosis of alcohol dependence. Nevertheless, many factors may additionally influence the level of BDNF. Moreover, it is indicated that differences in BDNF activity are gender-specific and are related to sex hormones [25]. In 2002, Solum and Handa [26] concluded that estrogen modulates *BDNF* expression in rats, and in 2012, Su et al. [27] showed that testosterone is a positive regulator of *BDNF* expression and progesterone increases BDNF production in the cerebral cortex explant and glial cells.

Studies examining the influence of the *BDNF* gene on alcohol addiction phenotypes have so far provided mixed results and have also reported no association [28]. A higher risk and earlier relapse after treatment were associated with the single nucleotide polymorphism in the *BDNF* gene i.e., the Val66Met (the Val/Val genotype) [29] In contrast, higher weekly alcohol consumption was reported in healthy met-allele carriers [30]. On the other hand, a recent study conducted on a large sample of alcohol-dependent patients did not show any significant association between *BDNF* Val66Met and alcohol dependence [28]. It seems that BDNF has a direct influence on the brain pathways typically engaged during the processing of rewards. A surge in BDNF levels in the ventral tegmental area (the region from which the dopaminergic neurons of the mesocorticolimbic dopamine system originate and extend to the dorsolateral and ventromedial striatum) triggers a state of reward similar to that produced by opiates in rats [31]. Furthermore, BDNF levels in the nucleus accumbens were found to be significantly lower in rats that prefer alcohol compared to those that do not [32]. The expression of *BDNF* in the striatum was found to hinder voluntary alcohol consumption in rats [33]. After acute ethanol intake or administration in mice, there was a more pronounced increase in *BDNF* expression in the dorsolateral striatum than in the dorsomedial striatum [34]. This suggests that exploring the relationship between the *BDNF* Val66Met genotype and individual variations in neuronal reward responses, as well as the role of orientation towards alcohol and its consumption, could provide valuable insights into the mechanisms required for alcohol dependence [35].

Alcohol dependence is a multifactorial disease, and one of potential factors associated with the development of AUD are personality traits. Personality is a dynamic organization of psychophysical systems that determines balanced behavior, perception, and thoughts about the environment and oneself, where personality traits are the dominant aspects [36]. For this reason, different personality models have been developed based on the expression of different personality traits. In the scientific community, the “Big Five” or “Five Factor Model” (FFM) currently obtains the greatest consensus [37,38]. This personality model is based on five main traits (neuroticism, extraversion, openness to experience, agreeableness, and conscientiousness) [39]. In FFM, neuroticism is associated with emotions experienced as unpleasant, such as anxiety, depression, and fear. Extraversion causes a person to perceive, construct, and experience events as stimulating and pleasant. Openness to experience is a dimension that is expressed in the spirit of openness, tolerance, and attraction to new things. Agreeableness considers relationships with others and their quality, unlike extraversion, which is more focused on the individual himself. Finally, conscientiousness enables the setting of goals and tasks and includes the ability to postpone the immediate gratification of desires and needs [40].

Bearing in mind that alcohol dependence depends on many factors, including sex, genetics, and personality, our study aims to analyze the association of the BDNF gene polymorphism rs6265 in women with alcohol use disorder and to analyze the association of personality traits with alcohol use disorder, as well as conduct multivariate analysis, i.e., analysis of the association of the rs6265 polymorphism of the BDNF gene and personality traits and alcohol dependency in women.

## 2. Results

The rs6265 variants distribution accorded with the Hardy–Weinberg equilibrium in the AUD and control groups (Table 1).

Significant difference in the distribution of *BDNF* rs6265 genotypes was revealed in the AUD subjects compared with the controls (G/A 0.19 vs. G/A 0.37; G/G 0.78 vs. G/G 0.61; A/A 0.03 vs. A/A 0.02, χ^2^ = 9.152, *p* = 0.0103). There was also a significant difference in the allele frequency of the analyzed polymorphism in the AUD group in comparison to the controls (G 0.88 vs. G 0.79; A 0.12 vs. A 0.21, χ^2^ = 5.091, *p* = 0.0240) (Table 2).

The means and standard deviations for the STAI trait and state scales and the NEO-FFI scales of the AUD and control subjects are presented in Table 3.

The AUD subjects, compared to the controls, scored higher scores in the assessment of the anxiety trait (7.49 vs. 4.90; Z = 7.284; *p* < 0.0001) and state (6.02 vs. 4.55; Z = 4.417; *p* < 0.0001) and NEO-FFI neuroticism (7.26 vs. 4.55; Z = 8.287; *p* < 0.0001) and openness (5.11 vs. 4.50; Z = 2.472; *p* =0.0134) scales. The AUD subjects, compared to the control group, scored lower on the extraversion (5.11 vs. 6.72; Z = −5.051; *p* < 0.0001), agreeability (3.84 vs. 5.49; Z = −5.380; *p* < 0.0001), and conscientiousness (4.97 vs. 6.88; Z = −5.844; *p* < 0.0001) NEO-FFI scales (Table 3).

The results of the 2 × 3 factorial ANOVA of the STAI and NEO five-factor inventory scales are summarized in Table 4.

### 2.1. STAI Trait Scale

The analysis revealed a significant effect of *BDNF* rs6265 genotype interaction and AUD, or lack thereof, on the STAI trait (F_2,207_ = 4.33; *p* = 0.0143; η^2^ = 0.040; Figure 1). The power for this factor was 75%, and ~4% was determined by the rs6265 and AUD or control on STAI trait scale score variance. Table 5 presents the post hoc test results. AUD patients with the G/A genotype had significantly higher STAI trait scale scores compared to controls with the G/A variant. AUD patients with G/G variant also had significantly higher STAI trait scale scores compared to the control group with the G/G genotype.


### 2.2. STAI State Scale

The analysis revealed a significant effect of *BDNF* rs6265 genotype interaction and AUD, or lack thereof, on the STAI state scale (F_2,207_ = 7.32 *p* = 0.00085; η^2^ = 0.066; Figure 2). The power for this factor was 93%, and ~7% was determined by the rs6265 and AUD or control on trait STAI state scale score variance. Table 5 presents the post hoc test results. AUD subjects with the G/A genotype had significantly higher scores for the anxiety state scale when compared to the controls who were G/A genotype carriers.

### 2.3. Neuroticism Scale

The analysis revealed a significant effect of *BDNF* rs6265 genotype interaction and AUD, or lack thereof, on the neuroticism scale (F_2,207_ = 3.17; *p* = 0.0440; η^2^ = 0.030; Figure 3). The power for this factor was 60%, and ~3% was explained by the rs6265 and AUD or control on neuroticism trait scale score variance. Table 6 presents the post hoc test results. AUD subjects with the G/A variant obtained significantly higher neuroticism scale scores compared to controls with the G/A genotype. Case subjects with the G/G variant also obtained significantly higher scores on the neuroticism scale scores compared to the control group with the G/G genotype.

### 2.4. Agreeability Scale

There was a significant effect of *BDNF* rs6265 genotype interaction and AUD, or lack thereof, on the agreeability scale (F_2,207_ = 3.10 *p* = 0.0473; η^2^ = 0.029; Figure 4). The power for this factor was 60%, and ~3% was explained by the rs6265 and AUD or control on the agreeability trait scale score variance. Table 6 presents the post hoc test results. AUD subjects with G/G genotypes had significantly higher Agreeability scale scores compared to controls with G/G genotypes. Case subjects with the A/A genotype also had significantly higher agreeability scale scores compared to controls with the A/A genotype.

## 3. Discussion

As the aim of this study, we chose a case-control analysis involving 101 women with alcohol use disorder. For the analysis, we selected the rs6265 single nucleotide polymorphism of brain-derived neurotrophic factor (BDNF), assessed personality with the NEO five-factor inventory, and assessed anxiety with the state-trait anxiety inventory. In addition, we calculated the interaction between *BDNF* genotypes, personality traits, and anxiety.

The first step of this study was to analyze the distribution of genotypes and alleles in the study group of alcohol-dependent women in comparison to the controls. We found a significant difference in the distribution of rs6265 genotypes between the alcohol-dependent and control groups. The G/G genotype occurred significantly more often in the group of alcohol-dependent women compared with the controls. We also found a significant difference in the distribution of rs6265 alleles between the study and control groups. Allele A was significantly more frequent in the control group compared with the study group.

Katsarou et al. [41] conducted a study to link selected SNPs, including rs6265 of the *BDNF* gene, to the development of alcohol dependence in a sample of the Caucasian population in the southeast. They noted that the A allele was associated with a reduced risk of alcohol dependence, while the GG genotype was associated with the risk of alcohol dependence. However, Nedic et al. [28] obtained different results. They conducted a study to assess the association between rs6265 and alcohol-related phenotypes among Caucasian patients. Their results, inversely to ours, do not show any statistical association between genotypes of rs6265 and alcohol dependence.

Su [42] conducted a study in which they investigated the association between three single nucleotide polymorphisms (SNPs) of the *BDNF* gene, including the rs6265 polymorphism analyzed in our study, and depression associated with excessive alcohol consumption (AD-D) and their possible effect on sertraline treatment. They studied 548 men, of whom 166 had AD-D and 312 were healthy controls. They showed that the A allele is over-represented in patients with AD-D compared to healthy subjects and AD-nD (excessive alcohol consumption without comorbid depression).

However, it is important to note that the rs6265 is not the only factor that can influence the development of alcohol dependence. Addiction is a complex disease that is influenced by many factors in addition to genetics, such as environment or personality. We already know that certain genetic variants can predispose a person to the development of alcohol dependence, but they are not the determining factor. It is hugely influenced by personality structure, which was examined and measured using the NEO five-factor inventory in our study.

In our study, we showed that alcohol-dependent women obtained significantly higher scores on the neuroticism and openness scales and lower scores on the extraversion, agreeableness, and conscientiousness scales compared to the control group. We also performed a multivariate analysis, which showed that alcohol-dependent women who were carriers of the *BDNF* G/A genotype had significantly higher scores on the neuroticism scale compared to the control group with the same polymorphic variant. Also, for the G/G variant, alcohol-dependent women had statistically significantly higher scores on the neuroticism scale compared to controls with the same variant. Our study also showed that alcohol-dependent women with the G/G genotype scored higher on the agreeableness scale compared to the control group with the same variant. The same is also true for the A/A genotype; alcohol-dependent women with this variant obtained significantly higher scores on the agreeableness scale compared to the control group with the same genotype. Other studied traits did not reveal significant associations.

Clark et al. [43] conducted a study on the general population and the influence of personality traits on alcohol consumption. They showed that high levels of neuroticism and low levels of conscientiousness were associated with earlier alcohol use, increased alcohol use, and alcohol problems. Furthermore, the Betkowska-Korpała [44] study suggested that neuroticism and conscientiousness may predict treatment outcomes. It was also observed that the abstinence maintenance group had higher agreeableness and conscientiousness, which was conducive to cooperating with others and undertaking and completing tasks. Furthermore, lower values of neuroticism components are associated with greater adaptability and greater participation in treatment than in the relapse group. In contrast, Finn and Robinson [45] confirmed the effect of neuroticism on AUD but reported no impact on other personality traits.

In addition, we analyzed anxiety as a trait and as a state using the STAI questionnaire. Compared to the control group, alcohol-dependent women showed significantly higher scores on anxiety as a trait and significantly higher scores on anxiety as a state. We also performed a multivariate analysis, which showed that women in the study group with the G/A genotype had significantly higher scores on the anxiety trait scale than the control group with the G/A genotype. The same was true for alcohol-dependent women with the G/G variant. Women with alcohol use disorder with this variant had significantly higher scores for the STAI trait scale than the control group with the G/G genotype. In contrast, analysis of anxiety as a state showed that AUD women with the G/A variant scored significantly higher for anxiety as a state compared to the control group with the G/A variant.

## 4. Materials and Methods

### 4.1. Participants

The study group consisted of 213 female volunteers, 101 with alcohol use disorder (AUD) (mean age = 45.74, SD = 11.11), and 112 were not dependent on any substance or behavior (mean age = 45.32, SD = 10.19). The Bioethical Committee of the District Medical Council in Zielona Góra (KB-07/72/2017) approved this study. All participants provided written informed consent before participating in the study. The study was conducted at the Pomeranian Medical University in Szczecin, in the independent laboratory of behavioral genetics and epigenetics. Alcohol-dependent subjects were recruited in the addiction treatment facility after at least 3 months of sobriety. The necessity for a minimum of three months of sobriety is predicated upon the requirement to stabilize the physical, emotional, cognitive, and executive functions in AUD individuals. These functions are necessary for the accurate completion of psychometric questionnaires. Both the AUD and the control group individuals were volunteers examined by a psychiatrist using the mini international neuropsychiatric interview (MINI), state-trait anxiety inventory (STAI) and NEO five-factor personality inventory (NEO-FFI).

### 4.2. Psychometric Tests

The MINI-international neuropsychiatric interview is a systematically organized interview process aimed at assessing the diagnoses of psychiatric patients based on the criteria set forth by DSM-IV and ICD-10.

The NEO five-factor inventory (NEO-FFI) encompasses these six components for each of the five traits: agreeableness, (straightforwardness, compliance, modesty, altruism, tenderness, trust), extroversion (warmth, emotion seeking, sociability, activity, positive emotions, assertiveness), neuroticism (susceptibility to stress, hostility, depression, self-awareness, impulsivity, anxiety), conscientiousness (self-discipline, competence, duty, consideration, order, striving for achievements), and openness to experience (values, aesthetics, actions, feelings, ideas, fantasy) [22].

The state-trait anxiety inventory (STAI) is an instrument utilized to quantify anxiety. It interprets anxiety as either a temporary state that is influenced by the individual’s situation or a relatively stable characteristic of the personality. The STAI is composed of two subscales, subscale X-1 analyses state anxiety and subscale X-2 analyses trait anxiety. The questions for both subscales are located on both sides of a single test sheet. The subscale comprises 20 items, and the respondent must select 1 of 4 categorized responses [46].

The transformation of the raw score to the sten scale was performed in accordance with the Polish standards for adults. It was presumed that a sten of 1–2 signifies very low results, 3–4 represents low results, 5–6 indicates average results, 7–8 denotes high results, and a sten of 9–10 is indicative of very high results.

### 4.3. Genotyping

The gDNA was purified from venous blood using standard procedures. Genotyping was conducted with the real-time PCR method. The fluorescence signal was graphed in relation to temperature, resulting in melting curves for each sample. *BDNF* rs6265 allele peaks were read at 56.94 °C for the G allele and 62.83 °C for the A allele.

### 4.4. Statistical Analysis

The agreement between the genotype frequencies and the Hardy–Weinberg equilibrium was examined using the HWE software (https://wpcalc.com/en/equilibrium-hardy-weinberg/, accessed on 5 April 2023). *BDNF* rs6265 genotype frequencies in AUD group and controls were tested using the chi-square test. The STAI (trait, state) and NEO five-factor inventory (neuroticism, conscientiousness extraversion, openness, and agreeability) subscales were compared between the analyzed groups using the Mann–Whitney U test. The homogeneity of variance condition was fulfilled (Levene test *p* > 0.05). The analyzed variables were not distributed normally. The relationships between *BDNF* rs6265 variants of AUD and control subjects, the STAI trait and state scales, and the NEO five-factor inventory were analyzed using a multivariate analysis of factor effects ANOVA [STAI/NEO-FFI scale × genetic feature × control and AUD × (genetic feature × control and AUD)]. STATISTICA 13 (Tibco Software Inc., Palo Alto, CA, USA) for Windows (Microsoft Corporation, Redmond, WA, USA) was used for the analysis.

## 5. Conclusions

Our study suggests that characteristics like anxiety (both as a trait and a state) and neuroticism could have a significant impact on the mechanism of substance dependency, particularly in females who are genetically susceptible. This is regardless of the reward system that is implicated in the emotional disruptions accompanying anxiety and depression.

## Figures and Tables

**Figure 1 ijms-25-06448-f001:**
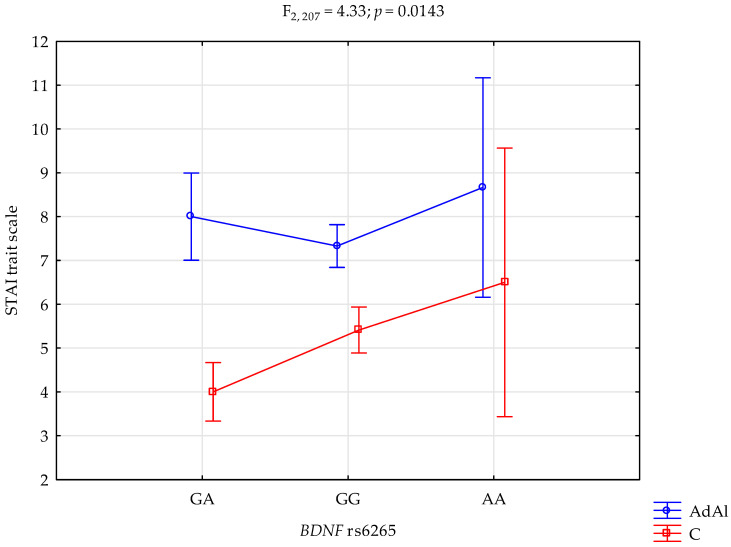
Interaction between the alcohol use disorder subjects (AdAl), controls (C), and *BDNF* rs6265 and STAI trait scale.

**Figure 2 ijms-25-06448-f002:**
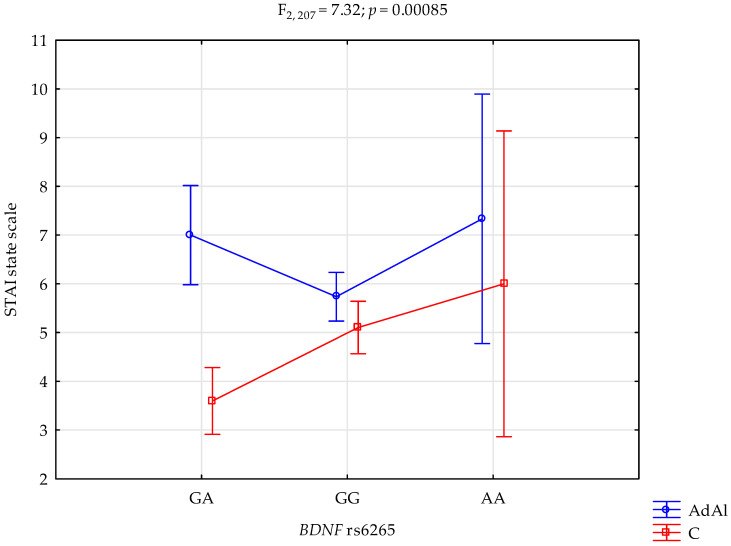
Interaction of the alcohol use disorder subjects (AdAl), controls (C) and *BDNF* rs6265 and STAI state scale.

**Figure 3 ijms-25-06448-f003:**
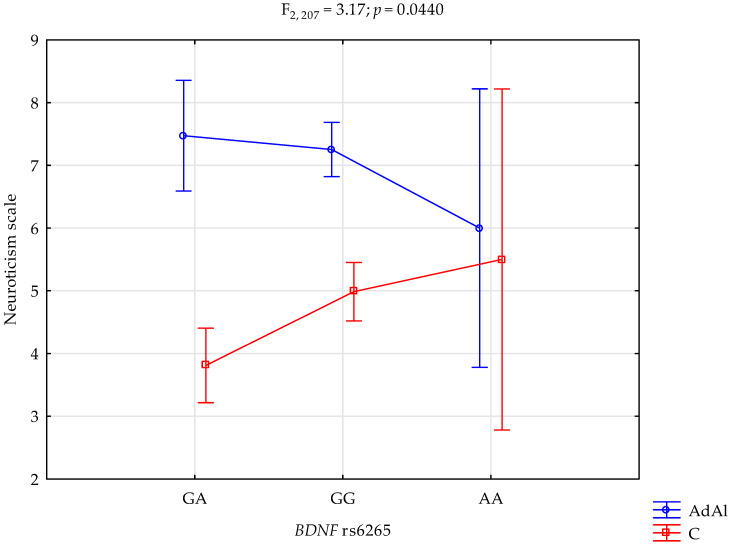
Interaction of the alcohol use disorder subjects (AdAl), controls (C), and *BDNF* rs6265 and neuroticism scale.

**Figure 4 ijms-25-06448-f004:**
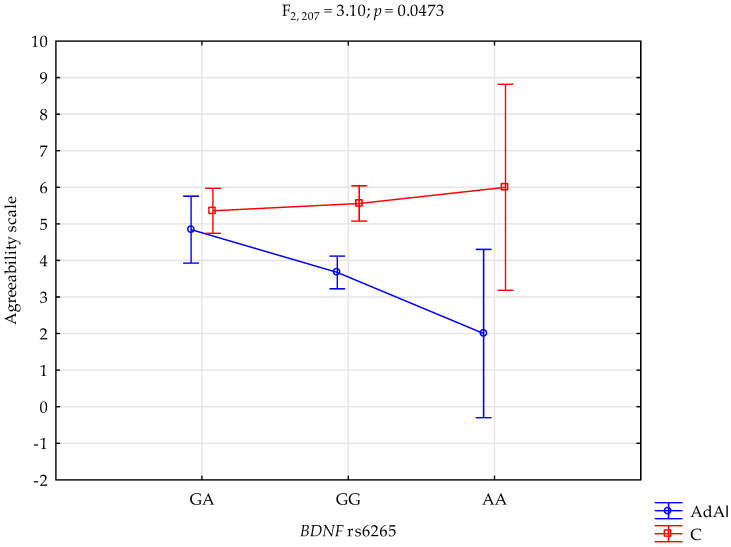
Interaction of the alcohol use disorder subjects (AdAl), controls (C), and *BDNF* rs6265 and agreeability scale.

**Table 1 ijms-25-06448-t001:** Hardy–Weinberg equilibrium for AUD and control subjects.

Hardy-Weinberg Equilibrium, Including Analysis for Ascertainment Bias		Observed (Expected)	Allele Freq	χ^2^ (*p*-Value)
*BDNF* rs6265				
AUD subjects *n* = 101	G/A	19 (21.9)	p (A)= 0.88 q (G)= 0.12	1.777 (0.1824)
	G/G	79 (77.5)		
	A/A	3 (1.5)		
Control subjects *n* = 112	G/A	42 (36.6)	p (A)= 0.79 q (G)= 0.21	2.486 (0.1148)
	G/G	68 (70.7)		
	A/A	2 (4.7)		

*p*-value—statistical significance, χ^2^—Chi^2^ test, *n*—number of subjects.

**Table 2 ijms-25-06448-t002:** Genotype and allele frequency of the *BDNF* gene rs6265 polymorphisms in the AUD and control subjects.

*BDNF* rs6265					
	Genotypes			Alleles	
	G/A *n* (%)	G/G *n* (%)	A/A *n* (%)	G *n* (%)	A *n* (%)
AUD *n* = 101	19 (18.81%)	79 (78.22%)	3 (2.97%)	177 (87.62%)	25 (12.38%)
Control *n* = 112	42 (37.50%)	68 (60.71%)	2 (1.79%)	178 (79.46%)	52 (20.54%)
χ^2^ (p-value)	9.152 (0.0103 *)			5.091 (0.0240 *)	

*p*-value—statistical significance, χ^2^—Chi^2^ test, *n*—number of subjects, *—statistical significance.

**Table 3 ijms-25-06448-t003:** STAI and NEO-FFI sten scores in AUD and control group.

State-Trait Anxiety Inventory NEO Five-Factor Inventory	AUD (*n* = 101)	Control (*n* = 112)	Z	(*p*-Value)
Anxiety trait	7.49 ± 2.20	4.90 ± 2.30	7.284	0.0000 *
Anxiety state	6.02 ± 2.51	4.55 ± 2.15	4.417	0.0000 *
Neuroticism	7.26 ± 1.86	4.55 ± 2.09	8.287	0.0000 *
Extraversion	5.11 ± 2.24	6.72 ± 1.91	−5.051	0.0000 *
Openness	5.11 ± 2.14	4.50 ± 1.66	2.472	0.0134 *
Agreeability	3.84 ± 1.87	5.49 ± 2.19	−5.380	0.0000 *
Conscientiousness	4.97 ± 2.33	6.88 ± 2.00	−5.844	0.0000 *

* statistically significant differences, *n*—number of subjects, Z—test statistics, M ± SD—mean ± standard deviation, *p*-value—statistical significance with Mann–Whitney U test.

**Table 4 ijms-25-06448-t004:** The 2 × 3 factorial ANOVA results for AUD and control subjects, NEO five-factor inventory, state-trait anxiety inventory, and *BDNF* rs6265.

STAI NEO FFI	Group	*BDNF* rs6265				ANOVA		
		G/A *n* = 61 M ± SD	G/G *n* = 147 M ± SD	A/A *n* = 5 M ± SD	Factor	F (*p*-Value)	ɳ^2^	Power (Alpha = 0.05)
STAI trait scale	Alcohol use disorder (AUD); *n* = 101	8.00 ± 1.45	7.33 ± 2.35	8.67 ± 1.15	AUD/control *BDNF* rs6265 AUD/control x *BDNF* rs6265	F_1,207_ = 14.42 (*p* = 0.0001) * F_2,207_ = 1.38 (*p* = 0.2541) F_2,207_ = 4.33 (*p* = 0.0143) *	0.065 0.013 0.040	0.966 0.294 0.747
	Control; *n* = 112	4.00 ± 2.24	5.41 ± 2.19	6.50 ± 0.71				
STAI state scale	Alcohol use disorder (AUD); *n* = 101	7.00 ± 1.97	5.73 ± 2.59	7.33 ± 1.15	AUD/control *BDNF* rs6265 AUD/control x *BDNF* rs6265	F_1,207_ = 6.07 (*p* < 0.0145) * F_2,207_ = 0.81 (*p* = 0.4447) F_2,207_ = 7.32 (*p* = 0.00085) *	0.029 0.008 0.066	0.689 0.188 0.935
	Control; *n* = 112	3.59 ± 1.96	5.10 ± 2.09	6.00 ± 0.71				
Neuroticism scale	Alcohol use disorder (AUD); *n* = 101	7.47 ± 1.39	7.25 ± 1.96	6.00 ± 1.73	AUD/control *BDNF* rs6265 AUD/control x *BDNF* rs6265	F_1,207_ = 11.60 (*p* < 0.0008) * F_2,207_ = 1.19 (*p* = 0.3058) F_2,207_ = 3.17 (*p* = 0.0440) *	0.053 0.011 0.030	0.924 0.259 0.603
	Control; *n* = 112	3.81 ± 1.80	4.99 ± 2.17	5.50 ± 0.71				
Extraversion scale	Alcohol use disorder (AUD); *n* = 101	5.53 ± 2.34	4.94 ± 2.16	7.00 ± 3.46	AUD/control *BDNF* rs6265 AUD/control x *BDNF* rs6265	F_1,207_ = 0.54 (*p* = 0.4618) * F_2,207_ = 0.22 (*p* = 0.8013) F_2,207_ = 2.08 (*p* = 0.1272)	0.003 0.002 0.020	0.114 0.084 0.425
	Control; *n* = 112	6.62 ± 1.74	6.82 ± 2.02	5.50 ± 2.12				
Openness scale	Alcohol use disorder (AUD); *n* = 101	4.79 ± 1.81	5.25 ± 2.20	3.33 ± 2.08	AUD/control *BDNF* rs6265 AUD/control x *BDNF* rs6265	F_1,207_ = 0.003 (*p* = 0.9503) F_2,207_ = 0.86 (*p* = 0.4248) F_2,207_ = 0.76 (*p* = 0.4692)	0.00001 0.008 0.007	0.050 0.197 0.178
	Control; *n* = 112	4.48 ± 1.58	4.51 ± 1.74	4.50 ± 0.71				
Agreeability scale	Alcohol use disorder (AUD); *n* = 101	4.84 ± 1.42	3.67 ± 1.89	2.00 ± 1.00	AUD/control *BDNF* rs6265 AUD/control x *BDNF* rs6265	F_1,207_ = 10.70 (*p* = 0.0013) F_2,207_ = 1.42 (*p* = 0.2432) F_2,207_ = 3.10 (*p* = 0.0473) *	0.049 0.014 0.029	0.902 0.303 0.592
	Control; *n* = 112	5.36 ± 2.33	5.56 ± 2.13	6.00 ± 1.41				
Conscientiousness scale	Alcohol use disorder (AUD); *n* = 101	5.16 ± 2.17	4.92 ± 2.41	5.00 ± 1.73	AUD/control *BDNF* rs6265 AUD/control x *BDNF* rs6265	F_1,207_ = 3.44 (*p* = 0.0649) F_2,207_ = 1.12 (*p* = 0.3276) F_2,207_ = 0.57 (*p* = 0.5690)	0.016 0.011 0.005	0.455 0.246 0.143
	Control; *n* = 112	7.29 ± 1.77	6.69 ± 2.11	5.00 ± 1.41				

*—significant result; M ± SD—mean ± standard deviation.

**Table 5 ijms-25-06448-t005:** Post hoc (Least Significant Difference) analysis of interactions of the AUD subjects, controls, and *BDNF* rs6265 and STAI state and trait scales.

***BDNF* rs6265 and Anxiety Trait Scale**						
	**{1}** **M = 8.00**	**{2}** **M = 7.33**	**{3}** **M = 8.67**	**{4}** **M = 4.00**	**{5}** **M = 5.41**	**{6}** **M = 6.50**
AUD subjects G/A {1}		0.2339	0.6261	0.0000 *	0.0000 *	0.3599
AUD subjects G/G {2}			0.3023	0.0000 *	0.0000 *	0.5991
AUD subjects A/A {3}				0.0005 *	0.0129 *	0.2817
Control G/A {4}					0.0013 *	0.1178
Control G/G {5}						0.4911
Control A/A {6}						
***BDNF* rs6265 and Anxiety State Scale**						
	**{1}** **M = 7.00**	**{2}** **M = 5.73**	**{3}** **M = 7.33**	**{4}** **M = 3.60**	**{5}** **M = 5.10**	**{6}** **M = 6.00**
AUD subjects G/A {1}		0.0288 *	0.8118	0.0000 *	0.0014 *	0.5507
AUD subjects G/G {2}			0.2284	0.0000 *	0.0915	0.8691
AUD subjects A/A {3}				0.0059 *	0.0945	0.5171
Control G/A {4}					0.0008 *	0.1414
Control G/G {5}						0.5791
Control A/A {6}						

M—mean, *—significant statistical differences.

**Table 6 ijms-25-06448-t006:** Post hoc (least significant difference) analysis of interactions of the AUD subjects, controls, and *BDNF* rs6265 and the NEO-FFI neuroticism and agreeability scales.

***BDNF* rs6265 and Neuroticism Scale**						
	**{1}** **M = 7.47**	**{2}** **M = 7.25**	**{3}** **M = 6.00**	**{4}** **M = 3.81**	**{5}** **M = 4.99**	**{6}** **M = 5.50**
AUD subjects G/A {1}		0.6586	0.2253	0.0000 *	0.0000 *	0.1750
AUD subjects G/G {2}			0.2760	0.0000 *	0.0000 *	0.2108
AUD subjects A/A {3}				0.0616	0.3789	0.7792
Control G/A {4}					0.0024 *	0.2325
Control G/G {5}						0.7134
Control A/A {6}						
***BDNF* rs6265 and Agreeability Scale**						
	**{1}** **M = 4.84**	**{2}** **M = 3.67**	**{3}** **M = 2.00**	**{4}** **M = 5.36**	**{5}** **M = 5.56**	**{6}** **M = 6.00**
AUD subjects G/A {1}		0.0244 *	0.0247 *	0.3580	0.1734	0.4420
AUD subjects G/G {2}			0.1616	0.0000 *	0.0000 *	0.1092
AUD subjects A/A {3}				0.0060 *	0.0032 *	0.0314 *
Control G/A {4}					0.6118	0.6609
Control G/G {5}						0.7613
Control A/A {6}						

*—significant statistical differences, M—mean.

## Data Availability

The genotyping and psychometric test results are available upon request.

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
