# Peer review of "Analysis of the BDNF Gene rs6265 Polymorphism in a Group of Women with Alcohol Use Disorder, Taking into Account Personality Traits"

_ijms, 2024, doi:10.3390/ijms25126448_

Round 1

Reviewer 1 Report

Comments and Suggestions for Authors

The paper is well-written, and I have some minor suggestions.

1. Page 1 of 15

Abstract

Line number: 29

Please spell out AUD

2. Page 2 of 15

Introduction.

The authors may need to add some prevalence data of AUD in women.

3. Page 3 of 15

Line number: 119

Results

The Results were placed in the second section. The authors may need to move them to the third section and replace the current Results section with the Materials and Methods section.

4. Page 12 of 15

Line number : 285-286

"The Bioethical 285 Committee of ..." This sentence seems incomplete; please add "approved this study" at the end of the sentence.

5.Page 12 of 15

Line number : 289

"at least 3 months of sobriety"

It is good to recruit AUD subjects who have been sober for 3 months. It would be better if the authors could justify why alcohol-dependent subjects need to be sober for at least 3 months to be recruited, as some readers may not have the related background.

6.Page 12 of 15

Line number : 311,314

"sten"

Is this a typo or is it "sten"?

7. Page 4 of 15

Line number: 141

"p=0134"

The author may need to revise it as p=0.0134 or p=.0134

Comments on the Quality of English Language

English is good.

Author Response

We sincerely thank you for all your valuable comments. Below is the location of all changes, which can additionally be seen in the changes tracking panel.

Comment 1.

Page 1 of 15

Abstract

Line number: 29

Please spell out AUD

Thank you for this suggestion. The AUD acronym was described.

Comment 2.

Page 2 of 15

Introduction.

The authors may need to add some prevalence data of AUD in women.

Thank you for this suggestion. The information has been added in lines 47-57.

Comment 3.

Page 3 of 15

Line number: 119

Results

The Results were placed in the second section. The authors may need to move them to the third section and replace the current Results section with the Materials and Methods section.

The article was written according to the instructions for authors given on the journal's website. The Research Manuscript template is as follows: Introduction, Results, Discussion, Materials and Methods, and Conclusions. Hence, the authors cannot change the order of sections.

Comment 4.

 Page 12 of 15

Line number: 285-286

"The Bioethical 285 Committee of ..." This sentence seems incomplete; please add "approved this study" at the end of the sentence.

Thank you for spotting the editing error. It has been corrected in lines 296-297.

Comment 5.

Page 12 of 15

Line number : 289

"at least 3 months of sobriety"

It is good to recruit AUD subjects who have been sober for 3 months. It would be better if the authors could justify why alcohol-dependent subjects need to be sober for at least 3 months to be recruited, as some readers may not have the related background.

 Thank you for the suggestion. The information has been added in lines 301-305

Comment 6.

Page 12 of 15

Line number : 311,314

"sten"

Is this a typo or is it "sten"?

Thank you for this question. “sten” is not a typo. Sten scale is a psychological test scale standardised so that the population mean is 5.5 and the standard deviation is 2, with 10 units in the scale.

Comment 7.

Page 4 of 15

Line number: 141

"p=0134"

The author may need to revise it as p=0.0134 or p=.0134

Thank you for spotting the editing error. It has been corrected.

Reviewer 2 Report

Comments and Suggestions for Authors

The manuscript is very interesting and raises the question of the relationship between genetic aspects and personality traits in women with alcohol dependence. Identifying specific individual characteristics that can influence the desire for alcohol is the most original aspect that could open new horizons for therapeutic and preventive interventions.

I found a limit in defining some emotions as "negative" (line 105 for example), I would have preferred the authors to refer to "emotions experienced as unpleasant" when writing about anxiety and depression. Would it be possible for the authors to revise this part?

I found the outline of the manuscript a bit confusing: inserting the results first and then the methodology makes it difficult to read a very interesting work. Would it be possible to move paragraph 4 immediately after the introduction?

Would it be possible to describe the Participants paragraph better? How was the control group enrolled? All the women are volunteers and the alcohol-dependent women were enrolled in the addiction treatment facility, it is not clear how the women in the control group were enrolled.

Regarding the results, would it be possible to include information relating to sociodemographic variables? How do the authors interpret the influence of environmental context variables? I would suggest, if space allows, to include a reflection by the authors on the complex interaction between genetics, personality traits and environmental variables.

The discussion and conclusions are clear and consistent with what the authors hypothesized and the bibliographical references are appropriate.

In summary, I would suggest the authors review the outline of the manuscript to help the reader understand this interesting work, original in its hypotheses and results. It would be helpful to include Materials and Methods after the introduction, followed by the results, discussion and conclusion

Author Response

We sincerely thank you for all your great job and very valuable comments.

Below is the location of all changes, which can additionally be seen in the changes tracking panel.

Comments:

  • I found a limit in defining some emotions as "negative" (line 105 for example), I would have preferred the authors to refer to "emotions experienced as unpleasant" when writing about anxiety and depression. Would it be possible for the authors to revise this part?

Thank you for this insightful comment. The authors absolutely agree with the reviewer and the change has been made in line 117

  • I found the outline of the manuscript a bit confusing: inserting the results first and then the methodology makes it difficult to read a very interesting work. Would it be possible to move paragraph 4 immediately after the introduction?

The article was written according to the instructions for authors given on the journal's website. The Research Manuscript template is as follows: Introduction, Results, Discussion, Materials and Methods, and Conclusions. Hence, the authors cannot change the order of sections.

  • Would it be possible to describe the Participants paragraph better? How was the control group enrolled? All the women are volunteers and the alcohol-dependent women were enrolled in the addiction treatment facility, it is not clear how the women in the control group were enrolled.

Thank you for the suggestion. The information regarding both study groups has been added in lines 301-305.

  • Regarding the results, would it be possible to include information relating to sociodemographic variables? How do the authors interpret the influence of environmental context variables? I would suggest, if space allows, to include a reflection by the authors on the complex interaction between genetics, personality traits and environmental variables.

Thank you for this question. It is regrettable that the full socioeconomic and sociocultural questionnaire involving environmental factors was not performed, and all the data pertaining to both the patient and control group are presented in the paper. Unfortunately, recruiting females with AUD is far more challenging due to their lower willingness to cooperate with researchers than men (clinical experience in Poland). Hence, the ability to gather data is significantly decreased. The Reviewer poses a substantiated comment. However, due to a lack of data, the authors decided not to discuss this aspect of AUD and keep this article part strictly related to the obtained results.

  • In summary, I would suggest the authors review the outline of the manuscript to help the reader understand this interesting work, original in its hypotheses and results. It would be helpful to include Materials and Methods after the introduction, followed by the results, discussion and conclusion

Thank you for all the valuable comments and suggestions.